# The Profile of the Foreign Investor in the Romanian Chemical Industry

**Iulia Iuga [1],\*, Aniela Danciu [2] and Imola Drigă [3]**

[1]    Department of Finance, "1 Decembrie 1918" University of Alba Iulia, 510009 Alba Iulia, Romania
[2]    Department of Statistics and Econometrics, Bucharest Academy of Economic Studies, 010374 Bucharest, Romania; aniela.danciu@csie.ase.ro
[3]    Department of Economics, University of Petroşani, 332006 Petroşani, Romania; imola.driga@gmail.com
\*    Correspondence: iuga_iulia@uab.ro

**Abstract:** The main aim of this study is to build the investor's profile in the Romanian chemical industry and to highlight the factors that influenced the decision of investing in Romania rather than other Central Eastern European countries. The data collection was performed in June 2019 and the list of the 150 foreign companies from the chemical industry was obtained from The National Trade Register Office. Data used in this research were collected using a questionnaire. Dependent variable represents the probability of investing in Romania, with the option of the other Central and Eastern European countries as reference group. The main part of our analysis focus on this question: "Which were the reasons that made you decide invest in Romania?" For analysis, a number of six main classes are used: Infrastructure, labor force, Agglomeration factors, Knowledge, Market Size and Cost factors (as independent variables). Main results consist in the presence of three factors with a positive impact. The paper also highlights that the main advantage considered by a foreign investor in Romania is represented by the cheap labor force. As a secondary conclusion, companies are also interested in other factors that are mentioned in the paper.

**Keywords:** chemical industry; the investor profile; foreign direct investments (FDI); factor analysis

## 1. Introduction

The chemical industry was permanently considered a key sector for the economic development. The economic development of the states was influenced by the development of the society: From the Stone Age, the Bronze Age, and the Iron Age to the industrial revolution. The delimitation of the industrial revolutions was based in criteria related to progress: The used energy; the succession of generations of technologies (from those based on the human physical work to those based on mechanics, electronics, electro-technical devices, chemicals, biology and informatics). An industrial revolution is preceded and engaged by scientific, technical and organizational progress (Table 1). The economy built in the architecture of the first industrial revolutions was gradually replaced, in an accelerating rhythm, by the 4.0 industry; the information and computerized technologies made possible the emergency of the fourth industrial revolution, together with new placements in the secondary and tertiary sector and the appearance of a new industrial order.

The fact that the enterprises are ready to jump to the Industry 4.0 show the degree in which the SMMs built team of experts with the role of implementing the new technologies [1]. The chemical industry practically is at the base of all the economy sectors and its strategies have a direct impact on the aftermarket users of the chemical products. Worldwide in 2018, China dominated the global classification of the chemical's producers, with sales of 1198 billion euro, more than the sum of the UE 565 billion euro and the USA 468 billion euro (Figure 1).

**Table 1.** The industrial revolutions—background [2].

| Period | | Scientific, Technical, and Organizational Progress |
|---|---|---|
| The first revolution | At the middle of the 18th century | It was based on coal, metallurgy, textiles, and steam engine. |
| The second revolution | At the end of the 19th century | It has as fundaments the electricity, the mechanics, the petroleum, the chemistry, the telegraph, the telephone, the railroads and, the steam boats. |
| The third revolution | At the middle of the 20th century | It was started by the discovery of the semiconductors, especially the transistor, and was centered on electronics, telecommunications, informatics, audio-visual, and nuclear technics, and also on the development of robots, automatization, spatial technology bio-technology. |
| The fourth industrial revolution 4.0 | At the beginning of the 3rd millennium | It produced turmoil in the production process, due to the internet and the technologies for the processing, and transmission of the information. |

From the global total of chemical products sales, China holds a percentage of 37.20%, followed by EU (15.60%) and USA (13.41%). Until 2030, China will probably represent more than half of the global chemical production, while EU and USA will represent only a quarter of this production [3].

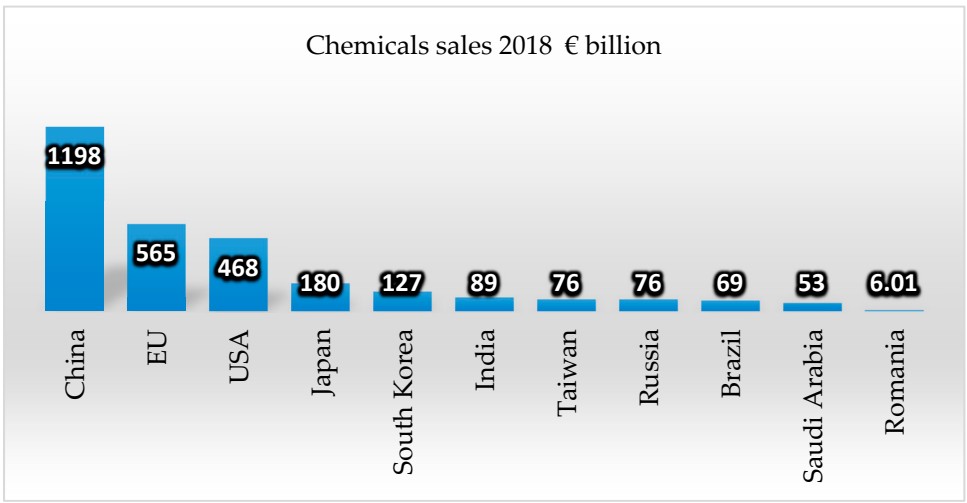

**Figure 1.** Top 10 sales by country. Authors' personal processing based on [4].

The European chemicals producers are confronted with a growing competition coming from the producers from other regions, which often have less strict rules, more favorable fiscal policies, and access to more inexpensive energy sources and raw materials. The field of interest for the chemical industry differs from one country to another. In some countries, the field of interest is strictly limited to chemical substances, according to NACE 20 from the Statistical classification of economic activities in the European Union. In other countries, the field of interest also covers the pharmaceutical sector and the rubber and plastics sector, NACE 21 and NACE 22. To the EU-28 level, there are 30,000 enterprises active in the chemical industry with a turnover of 565 billion EURO and 1,171,000 employees.

Figure 2 shows that to the EU-28 level, Germany and Poland have the most employees in the chemical industry and Poland has the biggest number of chemical companies, 11,000, five times more

than Germany. Related to the number of companies, a quarter of the countries have less than 330 companies (Q1): Estonia, Lithuania, Denmark, Austria, and Slovakia. Twenty-five percent of the countries have a number of companies between 330 (Q1) and 760 (Q2): Croatia, Latvia, Finland, Bulgaria, Netherlands, and Belgium. Twenty-five percent of the countries (Slovenia, Portugal, Romania, Greece, Czech Republic, and Germany) have a number of companies between 760 (Q2) and 2285 (Q3) —The Czech Republic and Germany have a number of companies almost double than the rest in their group. The last quarter belongs to the countries with over 2285 companies: Sweden, Italy, France, Spain, UK, and Poland.

Related to the number of employees in the chemical industry, 25% of the countries have less than 12,003 employees (Q1): Croatia, Latvia, Estonia, Lithuania, and Denmark; 25% of the countries have a number of employees between 12,003 and 39,500 (Q2): Slovenia, Portugal, Romania, Greece, Finland, and Bulgaria; 25% of the countries have a number of employees between 39,500 and 119,550 (Q3): Austria, Slovakia, Sweden, Netherlands, Belgium, and Italy, the last two countries with a number almost double compared to the countries from the same group. The last group includes countries with over 119,550 employees: Czech Republic, UK, France, Spain, Poland, and Germany. From this group, Germany is on the first place with 462,553 employees (Figures 2 and 3).

According to the situation presented in Figure 3, the first 25% of the countries, with the smallest turnover are: Estonia, Latvia, Croatia, Bulgaria, Lithuania and Slovakia. They had a turnover lower than 2.25 billion euro. The next quarter (Q2) includes the countries with a turnover in the chemical industry up to 13.264 billion euro: Greece, Romania, Portugal, Slovenia, and Denmark. The first 3 countries (Greece, Romania, and Portugal) registered a turnover less than a quarter from the maximum value of the turnover of the countries from this group. The Q3 quarter includes the countries with a turnover up to 55.5 billion euro: Austria, Czech Republic, Finland, Sweden, Poland, and Netherlands. This group also presents a powerful polarization of the countries; Poland and Netherland present values almost double than the rest of the countries. The last quarter includes the countries with a turnover higher than 55.5 billion euro: France, Spain, UK, Italy, Germany, and Belgium. Germany distances itself from the rest of the countries, with a turnover 3 times bigger than France, which is the second in hierarchy.

| | | Q1 | Q2 | Q3 | Q4 |
|---|---|---|---|---|---|
| **Number of companies** | Q4 | | | Sweden, Italy | France, Spain, UK, Poland |
| | Q3 | | Slovenia, Portugal, Romania, Greece | | Germany, Czech Republic |
| | Q2 | Croatia, Latvia | Finland, Bulgaria | Netherlands, Belgium | |
| | Q1 | Estonia, Lithuania, Denmark | | Austria, Slovakia | |
| | | Q1 | Q2 | Q3 | Q4 |
| | | **Number of employees** | | | |

**Figure 2.** The classification of the UE countries regarding the chemical industry in 2018, depending on the Number of companies and the Number of employees. Authors' personal processing based on [4].

| | | Q1 | Q2 | Q3 | Q4 |
|---|---|---|---|---|---|
| Number of companies | Q4 | | | Sweden, Poland | France, Spain, UK, Italy |
| | Q3 | | Slovenia, Portugal, Romania, Greece | Czech Republic | Germany |
| | Q2 | Croatia, Latvia, Bulgaria | | Finland, Netherlands | Belgium |
| | Q1 | Estonia, Lithuania, Slovakia | Denmark | Austria | |
| | | Q1 | Q2 | Q3 | Q4 |
| | | Turnover | | | |

**Figure 3.** The classification of the EU countries regarding the chemical industry in 2018, depending on the Number of companies and the Turnover. Authors' personal processing based on [4].

From Figures 2 and 3 results that Romania occupies an intermediate position comparing the rest of the analyzed countries, due to foreign direct investments reduced volume in the chemical field. An important part of the literature related to the process of localization of the foreign direct investment shows the fact that the availability and the cost of the workforce, respectively the human capital, play an important role in the selection of the location of a foreign investment. The cost is a major preoccupation in the selection of the FDI location, reflecting for the most part the theory of the international division of labor. In this context, the high rates of the remuneration were often considered as acting as discouragement factor for the FDI fluxes [5,6], but there are also certain opposite results. The choice of the location is a critical, but also complex and multi-dimensional problem, because it influences the efficiency of the investing companies [7,8]. The choice of the location for the foreign direct investment is explained by other researchers, in different approaches, among which the majority are based on the reasons of the investment [9,10].

Several empirical studies prove that the companies extend abroad searching the access to technology and know-how, which are unavailable in their origin country, or the access on the market of the host country [11,12]. Certain host countries possess superior technical capacities, incorporated in the qualified workforce, in the design, fabrication or know-how capacity, or in the technological infrastructure. This argument is in accordance with point of view referring to the capacity of seeking the life cycle of the product, affirming that the decisions related to the localization of the production plants are influenced by the novelty of the product [10]. Due to the fact that the new products need such capacities, the access of these capacities further influence the decisions of the companies related to the location.

A substantial empirical literature is dedicated to examining the determining factors of foreign direct investment (FDI) to the eastern European countries. Stack, Ravishankar, and Pentecost sustained that important in attracting FDI to the eastern European countries is the quality of labor rather than the cost of labor. These countries possess a highly educated labor force, allowing them to benefit from the technology and knowledge transfer associated with FDI [13]. In recent literature there are very few studies that establish the investor profile in chemical industry.

We highlight in this paper the profile of the investor in the Romanian chemical industry and the factors that influenced the decision of investing in Romania rather than other Central Eastern European countries.

With this background, the paper is structured on 5 parts: The first part is a short introduction reflecting the importance of the chemical industry and presents the global and the EU chemical industry, then the classification of Romania compared to other EU countries, the second parts presents the Romanian chemical industry and the foreign direct investments in the chemical industry; the third part presents the study per se ("The analysis of the foreign investors' reasons for the investment in

the Romanian chemical industry"); the fourth presents the policies for the encouragement of the development of the Romanian chemical industry; and last part presents the conclusions of the paper.

## 2. Materials

### 2.1. The Romanian Chemical Industry

The Romanian chemical industry was very well developed during the communist regime and divided on three big branches: the chemistry of natural gas, the chemistry of oil and the chemistry of salt. The chemistry and the petro-chemistry were considered by the communist regime the flagpole of the modernization of the Romanian economy. Romania had a tradition in the production of oil and natural gas, and also specialists with an international acknowledged experience. The big refineries and chemical plants built at the end of the 60's and during the 70's benefited of an extremely favorable economic context, when the first oil shock from 1973 completely changed the global power politics. This period represented the glorious time of the Romanian chemical and petrochemical industry. The Romanian oil production reached its peak in 1976: 15.1 million tones/year, the fourth place in Europe, after Russia, UK and Norway.

Until 1990, important material and human resources were allocated to this industry. A percentage of 90% from the Romanian chemical industry was destroyed after 1989. Starting with 1990, Romania transitioned from a centralized economy, built on the excessive domination of the state property through specific regulations and institutions, to an economy based on the mechanisms and the institutions of the free market, which is more and more globalized. Therefore, the production of synthetic rubber, synthetic fiber, polystyrene, polyethylene, and pesticide, and the local market became a prisoner of the imports.

According to The Romanian National Institute of Statistics, the Romanian exports from 2017 had a value of 2.76 billion euro, while the imports had a value of 10 billion euro, generating a difference of 7.2 billion euro. The total trade deficit was over 13 billion, resulting that more than a half of the deficit is represented by the chemistry. The reduction of the production capacity and the competitiveness issues lead to the appearance of a negative balance in 2007, the equivalent of 56% from the total trade deficit or almost 4% from the Romanian GDP. According to Eurostat data (Code DS-018995), Romania registered an annual growing rhythm of 4.6% for the chemicals import from non-EU sources, for the period 2007–2017, and classifies the last as increase percentage of the exports for the same period and on the same relation (only 2%).

Related to the chemical industry, from the data for 2018, the place of Romania in the European classification is the following (Figure 3): it is in Q2 as turnover and in Q3 as number of active companies, to the same level with Slovenia, Portugal, and Greece, showing the predominance in the chemical industry of the small companies, with relatively low values of the turnover. Romania is together with Slovenia, Portugal and Greece in the interval Q1–Q2 as number of employees, and in Q2–Q3 as number of companies, showing a predominance of the companies with a low number of employees (Figure 2). These aspects show the existence of a critical area in our external trade. Without improving the situation and reversing the negative trend, due to the loss of competitiveness compared to the EU and non-EU countries, it will be very difficult to stop the chronic course of the minus from the trade, where the chemical products detain the majority percentage.

Romania holds a modest place from the point of view of the attracted foreign direct investments. Related to the repartition of the FDI balance on the main economic activities at 31 December 2018, according to the data collected from the website www.bnr.ro (data related to the Foreign direct investments in Romania in 2018): The processing industry recorded 25,032 million euro; within it, the second activity benefiting of foreign direct investments was the oil processing, chemicals, rubber and plastics (5175 million euro), representing 6.4% from the total FDI to the national level. The percentage of the exports of the FDI enterprises from the sector "oil processing, chemical, rubber, and plastic products" in the total economy is 8.5%, with a value of 5371 million euro. The percentage of the import

of the FDI enterprises from the sector "oil processing, chemical, rubber, and plastic products" in the total economy is 8.8%, with a value of 6918 million euro (Figure 4).

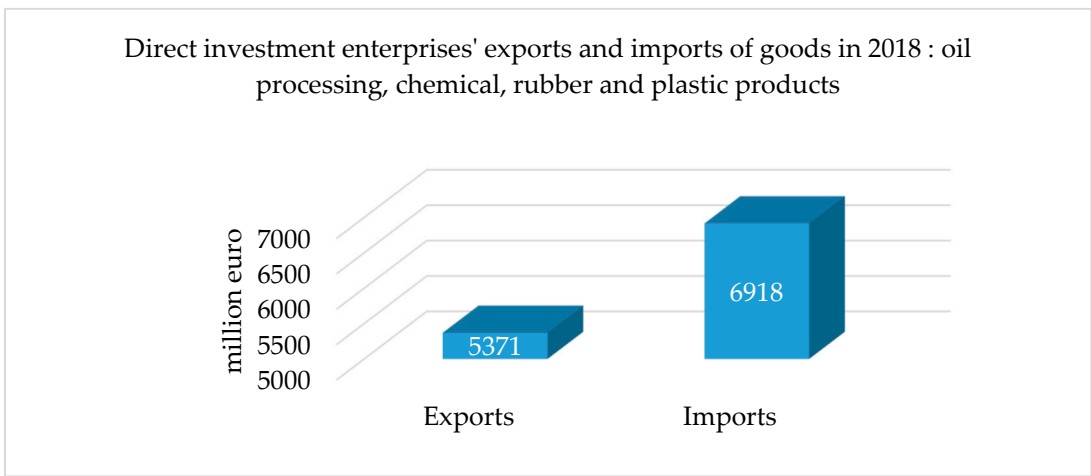

**Figure 4.** Romania—Direct investment enterprises' exports and imports of goods in 2018—oil processing, chemical, rubber, and plastic products. Authors' personal processing based on [14].

*2.2. Foreign Direct Investments (FDI) in the Chemical Industry*

On the global scene, the European chemical industry is facing challenging times. All the regions extend their chemical production. Almost a third of the EU investment in the chemical products is attributed to the petrochemical business. With a sum of 6.5 billion euro, the petro-chemistry is the largest investor in the EU chemical sector, followed by plastic materials and "industrial auxiliaries".

On the other hand, a change of paradigm in the request structures and the public preferences is developing from some time. The desire to use the resources in an efficient and ecological manner has notable effects on the energy consumption and the consumption habits. The trend toward the circular economy illustrates this transformation. The circular economy gains importance and the digitization lead to ample changes in all sectors.

These two basic subjects are of central importance from the trends in the chemical sector until 2030. The change of the public preferences for the durable production and consumption requires the development of new products and business models. In a circular economy, the chemical sector can use the growth potential: e.g., the support of the customers in reaching the sustainability objectives or the extension of the basic activity through new models of circular business, such as the chemical leasing. By developing strategies for satisfying the requests of the customers (requests that are continuously changing), the companies from the chemical industry bring an important contribution in reaching the UN sustainability objectives.

The choice of foreign investment location is a key strategic decision for internationalizing firms and has therefore received widespread attention from scholars in international business. Belderbos, Du and Slangen studied the distribution of investments across host countries including Romania. They studied control for the general attractiveness of global cities as foreign investment locations within a target country and conclude that subnational location choices for foreign investments are driven by a nuanced interplay between a country's contextual distance and investment and investor characteristics [15].

The great majority of the statistical surveys on FDI concentrated mostly on the developed countries, especially on the USA as host country [5,7,16]. The empirical studies suggested the following six sets of factors influencing the selection of the FDI: market factors, workforce factors, concentration factors, transportation infrastructure factors, technology access factors, and the governmental policy factors.

The literature clearly showed that the dimension and the increase of the market are important factors for the selection of the FDI placement [17–21]. A larger market offers a better opportunity for

the large-scale foreign investors, facilitating the sales not only on the internal market, but also on other markets. In their paper, Camarero, Montolio, and Tamarit analyzed the determinants of German stock outward FDI and find that determinants associated with horizontal FDI appear to be dominant for explaining FDI in developed countries while for the group of developing countries covariates associated with vertical FDI motives play a larger role [22].

Nakamura and Zhang considered models that describe foreign firms' and host countries' decisions on foreign direct investment (FDI) when host country product markets are characterized by certain types of market structures. They show that, under certain conditions, the host country and foreign parent firm (FP) are both better off in equilibrium if FP chooses to form a joint venture (JV) with a domestic partner in the host country, with some form of technology transfer [23].

Vo, Nguyen, Ho, and Nguyen examine determinants of foreign portfolio investment (FPI) from the developed countries to emerging economies using the new data from the IMF's large coordinated portfolio investment surveys and find that portfolio investors from G7 countries tend to depart from efficient portfolio allocation and they are biased towards some differentials between source and recipient countries [24].

Majeed and Ahmad [25] classified the studies on the human capital as determinant factor in the FDI attraction in two categories. The first category includes the studies covering the period 1960–1980, when the human capital had no significant importance on the FDI. The second category includes the studies analyzing the period 1980–2005, when the localization of the foreign investments became strongly dependent on the level of knowledge and the aptitudes of the workforce [5,26,27]. Studying the factors that determined the selection of location in the regions NUTS 2 in for countries in transition (Bulgaria, Romania, Poland and Hungary), for 4103 foreign investors from the manufacturing industry, Pusterla and Resmini [28] discovered that the investors operating in the low-tech sectors and branches prefer the regions with inexpensive workforce, while the companies operating in high-tech sectors and branches are attracted by the regions with qualified workforce.

Lu, Tao, and Zhu article uses an array of performance measures, including total factor productivity, exporting performance, wages, R&D investment, and firm survival, with one single data set to offer a fuller and more nuanced picture of the impact of FDI on domestic firms [29].

The effects of an agglomeration of companies prove to be important especially to the foreign companies, because a concentration of foreign companies show the potential of the location. Riedl [30], Du, Lu, and Tao [31], and Head and Ries [32] adopted the theory according to which the agglomeration has a positive and significant impact in the localization of the FDI. Managers had to learn to respect national path dependencies and specific skills of the local workforce [33].

The transport infrastructure refers to the availability of adequate highways, railways, ports, airports and other facilities aiming the operational efficiency. The studies confirmed that the FDI was attracted by the regions with a better transport infrastructure [16,19,31,34].

Finally, the government policies were in general perceived as a key variable in the modification of the FDI fluxes between the regions. In this respect, the first researches had the tendency to highlight the influence of the trade rates on the selection of the location [35]. Lunn [36], e.g., considered that the high trade barriers lead to high FDI levels from USA to the European countries. Other studies concentrated on other types of governmental policies, such as the corporate tax rates, the length of the fiscal vacations, and other financial stimulants [7,34]. With few exceptions [37], the research until the present time offers a general support to the idea that the promotion policy has a positive impact on the FDI fluxes.

## 3. The Analysis of the Foreign Investors' Reasons for the Investment in the Romanian Chemical Industry

The analysis of the foreign investors' reasons for the investment in the Romanian chemical industry as main purposes:

1.　　Building the profile of the foreign investor in the Romanian chemical industry;

2. highlighting the factors that influenced the decision to invest in Romania against the investment in other countries from the central or eastern Europe; and

3. offering, through our analysis, aggregated information that can be used by authorities for the establishment of Policies for the encouragement of the development of the chemical industry.

### 3.1. Data Collection and Sampling Design

The data for the present research are obtained from questionnaires, applied with the aim of examining the determinant factors of the FDI in Romania. The data collection was performed in June 2019. The list of the 150 foreign companies from the chemical industry was obtained from The National Trade Register Office and contained the names and the addresses of the foreign companies already established on the Romanian market, as FDI, before 2018 and functioning in 2018 in the chemical industry (19–22 NCEA), with more than 100 employees and a capital with more than 50% as foreign contribution.

Due to the fact that the list was one year old, we examined its validity by checking the contact data of each investor through internet and telephone. We discovered that 10 companies were not possible to be contacted anymore. From the rest of the companies, we obtained responses from 125 companies, which represent a good response rate.

The questionnaire solicited various types of information, from the general information (the establishment year, the origin, the number of employees, and the sales and turnover) to information specific to the localization in Romania (e.g., the market entry mode, the region, and the reasons of the investment).

To encourage the participation, we established together with the potential responders a system in three stages, using the internet and the telephone. During the first stage, before sending the questionnaire, we sent an introduction email to the superior manager of each company, after verifying the contact data. The aim of the email was to explain the objectives and the importance of the study and to solicit the participation of the company. During the second stage, the questionnaire was sent to the responders or the responders were contacted by phone for answering to the questionnaire. In the final stage, a letter of appreciation was sent to all the responders.

In order to give our results a greater reliability we conducted an analysis on the selection mechanism that generated our sample. The main goal of this analysis was to obtain evidences that this mechanism was quite similar to randomization. For all our targeted firms we have received from the authorities a few administrative data that gave us the possibility to construct three variables that we used as covariates in our analysis. The three variables are as follows: (1) Dichotomous variable—*technology level of the activity* (High Tech/Low Tech); (2) dichotomous variable—*EU membership of the investor* (EU member/ Non EU member); (3) ordinal three classes variable—*number of employees in 2018* (low number, medium number, large number).

We compared the distributions of these three variables for the sample of respondents with the distributions of the same variables in the initial target population and in the non-respondents' sample. We used the chi square and the Binomial tests from the SPSS software package to test the hypothesis that the distributions of the respondents for all three variables do not deviate significantly from their expected distributions (the expected distributions were constructed based on distributions from the target population).

We have no statistical evidence to assume that the obtained distributions (of the respondents) deviate significantly from the expected distributions (distribution of each variable in the population). The *Asymp. Sig (1-tailed)* values for the Binomial test for the first dichotomous variable was 0.27 and for the second one was 0.113 (higher than 0.05). For the last variable the chi square value was 0.28 (*Asymp. Sig 0.885*). All tests were performed with a 95% Confidence level.

Summarizing, still acting with caution, we can assume that the selection mechanism, the decision of participating to the survey, generated a sample pretty close to a random sample.

### 3.2. Selection of the Variables

### 3.2.1. Independent Variables

According to the theoretical results extracted from the literature, we selected 18 aspects measuring the importance of investment in a certain country. The main part of our analysis will focus on the sixth question (the complex question): "Which were the reasons that made you decide invest in Romania?" The six main classes mentioned before are as follows: Infrastructure, Labor force, Agglomeration factors, Knowledge, Market Size and Cost factors. The answer to all eighteen questions, included in the six clusters, is a scale with five values: 1—This factor was not taken in consideration, 2—Very little importance, 3—Little importance, 4—Important, 5—Very Important. (Table 2).

**Table 2.** Explanatory variables.

| Variables | Calculation |
|---|---|
| *Finfr* | Indicator obtained using the following indicators:<br>• The transport costs<br>• The traffic and road conditions<br>• The existence of airports/ports in the proximity<br>• The viability of the land<br>• Favorable conditions for distribution |
| *Fwork* | Indicator obtained using the following indicators:<br>• The existence of available workforce<br>• The low cost of the workforce<br>• The existence of qualified workforce<br>• The high educational level of the population |
| *Faglom* | Indicator obtained using the following indicators:<br>• The existence of suppliers of raw materials and other materials in the region<br>• The existence of companies with similar activity<br>• The existence of foreign companies in the region |
| *Fknow* | The score for the factor: the existence of universities or research centers in the region |
| *Fmarket* | The score for the factor: the existence of a good market for the products |
| *Fcost* | Indicator obtained using the following indicators:<br>• The low price of land or rentals<br>• The availability of raw materials to a low price<br>• Fiscal facilities offered in the region to the investors |
| DUM*90-06* | DUMMY: 0 in the investment started in 1990-2006, and 1for the contrary situation |
| DUM*ht* | DUMMY: 1 for high tech and 0 for low tech |

Using a multi-step method implying correlation analysis, factor analysis and several logistic regressions we have chosen the best construction for the involved independent variables. Each time Factor Analysis was performed using a PC Extraction method and a Varimax rotation method. Factor scores were created using the Regression Method (1) available in SPSS (the results were a little better with this refined method than with the simple weighted sum method (2)).

$$F = Z * R^{-1} * S \tag{1}$$

Z—Matrix of standardized observed variable scores
R—Correlation matrix for the observed variables
S—Factor structure matrix

$$F = Z * S \tag{2}$$

Because some of the independent variables were the initial items measured on a five point scale and some of them were constructed using combinations of the initial items (Factor scores) we have decided to standardize all of them before estimating the model.

The first class of factors "Infrastructure" (Table A1) includes five topics as follows: (1) Transportation costs, (2) quality of the roads, (3) the existences of the airports nearby, (4) the existence of viable land for the investment, and (5) favorable conditions for distribution of the products. Transportation costs were considered as being important and very important by about 66% of all investors. Also the existence of viable land for the investment was considered by 73.4% of the interviewed managers as being a crucial reason in the location choosing strategy. Concerning for the authorities might be the fact that almost 70% of the respondents do not consider the existence of airports nearby as being a factor that might require attention when locating an investment.

The first class of factors was included in the model using the estimated score for the Factor resulted from the Factor Analysis conducted on the class of Infrastructure related items. The new construct is correlated with the last two items (Table 3) and it might be regarded as a measure for *the viability of the land and favorable conditions for distribution*. The values of this independent variable are standardized.

**Table 3.** Infrastructure items Factor Analysis.

| Component matrix (Factor loadings) | 0.298 | 0.362 | 0.349 | 0.819 | 0,721 |
|---|---|---|---|---|---|
| Component score Coefficient Matrix | 0.142 | 0.198 | 0.182 | 0.503 | 0.476 |

The second class of factors "Labor force" (Table A1) includes 4 topics as follows: (1) The existence of available labor force, (2) the low cost of the labor force, (3) the existence of qualified labor force, (4) the high level of education of the population. Important to emphasize is that all four topics were considered a major factor in the process of strategic planning (important or extremely important) by almost 90% of the companies from our sample. The most important is the existence of available work force, followed by low cost of this workforce, by the existence of qualified workforce and by the high level of education.

The second class of factors was included in the model using the estimated score for the first Factor resulted from the Factor Analysis conducted on the class of Labor related items. The new construct is highly correlated with the last two items (Table 4) and it might be regarded as a measure for *the existence of qualified and highly educated workforce*. The values of this independent variable (the score) are standardized.

**Table 4.** Labor items Factor Analysis.

| Component matrix (Factor loadings) | 0.283 | −0.109 | 0.725 | 0.809 |
|---|---|---|---|---|
| Component score Coefficient Matrix | 0.121 | −0.178 | 0.547 | 0.513 |

The third class of factors "Agglomeration" was divided into three main topics: (1) The existence of suppliers in the region, (2) The existence of other companies with the same activity field in the region and (3) The existence of other foreign companies in the region. These factors are considered as being important by appreciatively about a third of the responding managers.

The third class of factors was included in the model using the estimated score for the Factor resulted from the Factor Analysis conducted on the class of Agglomeration related items. The new construct is correlated with all three initial items (Table 5) and it might be regarded as a general measure for Agglomeration related factors. The values of the independent variable included in the model resulted also through standardization.

**Table 5.** Agglomeration items Factor Analysis.

| Component matrix (Factor loadings) | 0.897 | 0.713 | 0.738 |
|---|---|---|---|
| **Component score Coefficient Matrix** | 0.603 | 0.586 | 0.497 |

The "Cost factors" was divided into three main topics: (1) tax incentives for investors, (2) low rent levels or low land acquisition price, (3) availability of cheap raw materials in the area. The low level of rents or the low land acquisition price and the general operating costs are considered as important determinants of the future investment by almost 80 % of the respondents. Tax incentives and the availability of raw materials are considered as being an important determinant for the made investment by almost 65% of the respondents.

This class of factors was included in the model using the estimated score for the Factor resulted from the Factor Analysis conducted on the class of Cost related items. The new construct is highly correlated with the first two items (Table 6) and it might be regarded as a measure for *the low price of land and the availability of raw materials.* The values of this independent variable are standardized.

**Table 6.** Cost items Factor Analysis.

| Component matrix (Factor loadings) | 0.767 | 0.870 | 0.172 |
|---|---|---|---|
| **Component score Coefficient Matrix** | 0.535 | 0.494 | 0.093 |

### 3.2.2. Control Variables

According to Altomonte [38], time dummy variables have a significant impact on the investment made in the Central and Eastern Europe by a multinational company. Therefore, the dummy variables were introduced to control the variation in time, appearing due to the economic changes specific to the Central and Eastern European countries. Pusterla and Resmini [28] affirm that the factors specific to the field of activity affects to option for localization of the multinational companies in the Central and Eastern European countries. Thus, the dummy variable for the type of industry of a specific company (high-tech or low-tech) was introduced. The classification of the activities of chemical production (Table 7) is based on the NACE code (statistical classification of economic activities in the European Community) and the number of the companies classified by type of activities (Table 8) is based on the author's processing, being presented in the tables below:

**Table 7.** The classification of the activities of chemical production [39].

| High-Tech | | Low-Tech | |
|---|---|---|---|
| **NACE Code** | **Title** | **NACE Code** | **Title** |
| 20 | Manufacture of chemicals and chemical products | 19 | Manufacture of coke and refined petroleum products |
| 21 | Manufacture of basic pharmaceutical products and pharmaceutical preparations | 22 | Manufacture of rubber and plastic products |

**Table 8.** Number of the companies by the type of activities.

| | High-Tech | | Low-Tech | |
|---|---|---|---|---|
| | **NACE Code 20** | **NACE Code 21** | **NACE Code 19** | **NACE Code 22** |
| **Number of companies** | 33 (26.4%) | 20 (16%) | 8 (6.4%) | 64 (51.2%) |
| **Average number of employees** | 486 | 299 | 1003 | 324 |
| **Number of EU investors** | 27 (81.81%) | 20 (100%) | 6 (75%) | 57 (89.06%) |

We notice that the low tech firms represent a majority, companies with a low technological production process choosing Romania for the low cost of the labor force. Although, the companies of Manufacture of coke and refined petroleum products are the least, they have the biggest average number of employees due to the existence of important petroleum companies, such as: RAFO SA, PETROTEL—LUKOIL SA and ROMPETROL RAFINARE SA.

Regarding the origin of the investors, we notice that in all types of activities there are mostly investors from the EU.

### 3.3. Regression Model

In order to assess the importance of each factor (item) in the location-choosing process we have used a methodology based on a logistic econometric modelling process. We have decided to express the binary answer (YES/NO) to the question, *"Was Romania the first option to locate your investment, at the beginning of your decision process?"*, as a function of individual items or items' combinations.

The model was estimated using SPSS at the level of the entire sample. The structural form of our model is listed below:

$$Pr(y = 1|x) = exp(z)/[1 + exp(z)], \tag{3}$$

where:

$$z = \beta 0 + \beta 1 Finfr + \beta 2 Fwork + \beta 3 Faglom + \beta 4 FACTcost +$$

$$+ \beta 5 Fknow + \beta 6 Fmarket + \beta 7 DUMan + \beta 8 DUMht$$

### 3.4. Empirical Results

We can observe the presence of three factors with a positive impact: the labor force (Fwork), the existence of universities and research centers (Fknow), the cost (Fcost), meaning that, Romania was preferred by the foreign investors rather than other Central and Eastern European countries (Table 9). It is important to underline that more than half of the investors in the chemical production are companies with a low level of production technology. This fact is a proof that the main advantages considered by a foreign investor in Romania are represented by the cheap labor force, the low price of land and the availability of raw materials. The investors prefer Romania due to the fact that the labor force is more qualified and more educated. An important reason for the localization of the investment was the existence of enterprises specialized in domains with a high degree of technology in certain regions of the country and with a qualified labor force (the chemical industry with tradition in Mureş County, Vâlcea County, Prahova County, and Iasi County).

**Table 9.** Models parameter estimation.

| Variable | Coefficients | Significance |
|----------|:------------:|:------------:|
| Constant | 1.197 | 0.000 |
| Finfr | 0.265 | 0.510 |
| Fwork | 0.162 | 0.034 |
| Faglom | –0.113 | 0.587 |
| Fcost | 0.029 | 0.026 |
| Fknow | 0.012 | 0.037 |
| Fmarket | –0.526 | 0.003 |
| DUMht | 0.815 | 0.850 |
| DUM*90-06* | 0.912 | 0.398 |

In the same time, the market factor (Fmarket) is a significant negative factor, meaning that, related to the market, Romania, compared to the other Central and Eastern European countries, is less attractive to the investors, despite the fact that Romania is one of the biggest countries in Central Eastern Europe (as population and surface) and it could be a great market place. We can observe that the start year is

not a significant factor, therefore Romania's accession to the European Union had no influence on the decision of investing in Romania.

## 4. Policies for the Encouragement of the Development of the Chemical Industry in Romania

The chemical industry must face profound technical, economic and social changes. By adopting an adequate industrial policy, competitive frame-conditions can be created for the chemical industry to a global level. The chemical industry should have sectoral policies (through state policies), and also company policies. According to this study, the main advantages considered by a foreign investor in Romania are represented by the cheap labor force, the low price of land and the availability of raw materials. The investors prefer Romania due to the fact that the labor force is more qualified and more educated.

Regarding empirical results presented, we propose four policies that should be implemented by the Romanian state:

- Investments in research-development—it is difficult to expect the achievement of the re-industrialization through innovation and intelligent processing, and, in general, the competitive development of the chemical industry, without investments in the research and development. Due to the fact that these investments are not possible to be supported by the small and medium-sized enterprises, the public sources and the European funds are essential for the industrial research and innovation;
- The public sector should support the development of necessary networks and the creation of cross-sectoral platforms and innovation groups for the exchange of know-how and for the availability of raw materials;
- Policies for the support of the re-industrialization. The re-industrialization represents the capitalization of the production potential based on the (existing) national raw materials, the available energetic resources and the oil and natural gases, which can be imported or from the national market. Nowadays, related to the chemical industry, Romania possesses the necessary raw materials (salt, petroleum, natural gas) for a re-industrialization based on modern technology and the economic principles of the world economy;
- The establishment of a well-developed system for the protection of the intellectual rights, especially of the chemical industry patents. Innovation is one of the major pillars of the policy for the chemical industry. The determinant factor of innovation is the research-development, concretized in industrial advantages.

## 5. Conclusions

The present study built the profile of the investor in the Romanian chemical industry and to highlight the factors that influenced the decision of investing in Romania rather than other Central Eastern European countries. It is important to underline that more than three quarters of the investors in the chemical production are companies with a low level of production technology. This fact is a proof that the main advantage considered by a foreign investor in Romania is represented by the cheap labor force. The companies are also interested by the available workforce, the existence in the region of suppliers of raw material and other materials, the existence in the region of universities or research centers, the low price for land or rentals, and the availability of raw materials with a low price. The decrease of the occupation level of the workforce in the field of the chemical industry coincided with the increase of the number of employees orienting toward high qualified occupations and with the increase of the number of employees with a tertiary education.

The present national regulations do not support too intensely the competitiveness of the chemical industry in Romania. A more active policy for the image, also supported by the state institutions, is essential for the Romanian chemical industry.

Finally, the study offers aggregated information that can be used by authorities for the encouragement of the chemical industry development.

**Author Contributions:** I.I. and A.D. conceptualization and supervision; I.I. and A.D. writing—original draft preparation, methodology; I.D. writing—review and editing. All authors have read and agreed to the published version of the manuscript.

**Funding:** This research received no external funding.

**Acknowledgments:** The authors gratefully acknowledge the constructive comments and suggestions made by the three anonymous reviewers.

**Conflicts of Interest:** The authors declare no conflict of interest.

## Appendix A

**Table A1.** $Q_6$. Which were the reasons for the investment in Romania?

| The Reason | Response Variants | | | | | Average Score |
|---|---|---|---|---|---|---|
| **Infrastructure** | | | | | | |
| The transportation costs | 1 | 2 | 3 | 4 | 5 | 4.11 |
| The good quality of the highways | 1 | 2 | 3 | 4 | 5 | 2.71 |
| The existence of airports in the proximity | 1 | 2 | 3 | 4 | 5 | 2.12 |
| The viability of the lands in the region (the insurance of utilities) | 1 | 2 | 3 | 4 | 5 | 4.21 |
| Geographical conditions favorable to distribution (the distance to the beneficiary) | 1 | 2 | 3 | 4 | 5 | 2.72 |
| **Workforce** | | | | | | |
| The existence of available workforce | 1 | 2 | 3 | 4 | 5 | 4.63 |
| The reduced cost of the workforce | 1 | 2 | 3 | 4 | 5 | 4.45 |
| The existence of qualified workforce | 1 | 2 | 3 | 4 | 5 | 4.13 |
| The high educational level of the population | 1 | 2 | 3 | 4 | 5 | 4.07 |
| **Agglomeration factors** | | | | | | |
| The existence of suppliers for raw materials and other materials in the region | 1 | 2 | 3 | 4 | 5 | 3.62 |
| The existence of companies with similar activity | 1 | 2 | 3 | 4 | 5 | 3.13 |
| The existence of foreign companies in the region | 1 | 2 | 3 | 4 | 5 | 2.15 |
| **Other factors** | | | | | | |
| Fiscal facilities for the investors | 1 | 2 | 3 | 4 | 5 | 3.07 |
| The existence of universities or research centres | 1 | 2 | 3 | 4 | 5 | 3.37 |
| The low price of land or rentals | 1 | 2 | 3 | 4 | 5 | 4.25 |
| The availability of inexpensive raw materials | 1 | 2 | 3 | 4 | 5 | 3.22 |
| The existence of a good market for the products | 1 | 2 | 3 | 4 | 5 | 2.78 |
| The operating cost of the company | 1 | 2 | 3 | 4 | 5 | 3.45 |

Note: 1—This factor was not taken in consideration, 2—Very little importance, 3—Little importance, 4—Important, 5—Very Important.

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
