# Peer review of "The Profile of the Foreign Investor in the Romanian Chemical Industry"

_processes, doi:10.3390/pr8030348_

Round 1

Reviewer 1 Report

After reading the manuscript, these are my comments:

1) The Introduction is excessively general. It mainly focuses on the historical evolution of the industrial sector and includes a too short reference to the objective of the study, methods and main results. Moreover, the research is poorly motivated.

2) Regarding Section 2, it seems to be concentrated almost exclusively on the European chemical industry without a proper review of the sector at a global scale. Besides, a part of the contents in this section should be moved to the Introduction in order to contextualize the research (specially for readers not familiar with the Romanian economy). Finally, the SWOT analysis in sub-section 2.5 is rather superfluous and out of context.

3) In Section 3, the model specification is unclear: in sub-section 3.2 a multinomial logit model is described, but estimation results shown in the Table 8 (together with the equation in sub-section 3.5 indicates that the specification considered is a logit model indeed. At this point, as the dependent variable is dichotomic (Romania vs Other countries) I don't see the need for considering a multinomial model instead of a much simpler logit or probit specification. It is also unclear which was the weighting scheme applied to indicators' scores in order to compute the explanatory variables such as Finfr, Fwork, Faglom, etc. With respect to the Empirical results, the authors seems to pay little attention to the statistical significance of the estimates: except for the Fmarket variable, statistical significance is achieved only at the 5% level, thus the results are rather little robust.

4) The section 4 is devoted to policy recommendations, notwithstanding, in general there is not a clear connection between empirical results presented in the previous section and the "long list" of policies included in the manuscript.

Reviewer 2 Report

SWOT: Where do you get the data to build the table 2 to presents the SWOT analysis for the localization in the Romanian chemical industry? for example, the Threats aspect

Reference: Most of them are historical literature wherein, lack the recent years reference data, meanwhile need to supplement from 2015 to 2019.

Introduction: You may combine the Materials into the Introduction.

Reviewer 3 Report

-Title: It is sufficient with the second phrase, “The profile of the foreign investor in the Romanian Chemical Industry”.

-Abstract needs more information about method and less about industrial context. Include the main results and conclusions.

-Keywords must be not related to the title.

-Source is missing in Tables 1 and 2.

-Figures 1 and 2 are redundant, select one of this. Include the Romanian percentage.

-Figures 3 and 4 could be unified.

-Some statistical information about companies would be good according to the main variables of questionnaire.

-What are the names of each company? A link with a list of companies would be necessary.

-Table 6 could include some statistical information. How many companies or percentage for each item?

-In conclusions the first paragraph could be eliminated.

-Annex 1 must be in methodological section, including some statistical information.

Round 2

Reviewer 1 Report

After reviewing the revised version of the manuscript I appreciate the authors have taken into account my comments and suggestions. However, I still have a concern regarding the specification of the logistic model in section 3.3.

Briefly, in the revised manuscript the econometric specification seems to correspond to a linear probability model (i.e. the 0/1 outcome of the choosing process is expressed as a linear function of the explanatory variables) instead to a logit model. In this regard, I suggest the model specification to be written as:

Pr(y=1|x) = exp(z)/[1+exp(z)],

where:

z = b0 + b1Finfr + b2Fwork + ... + b8DUMht.

Author Response

We really appreciate your comments and suggestions and we tried our best  making all the suggested modifications. We mentioned the model specification on  the paper as you suggested.

Reviewer 3 Report

The final result is satisfactory but this paper is a sectorial analysis related to a relatively minor country in the analyzed topic.

Probably a comparative analysis among countries could be better, perhaps at European level or East-European level.

Author Response

We really appreciate your comments and suggestions and we tried our best  making all the suggested modifications.

The main goal of this study is to identify the main determinants of the foreign direct investments in Romania in the chemical industry.

Despite the growing interest in the subject, to our knowledge, there is still no satisfactory empirical work which can explain the determinants of FDI flows in the Romanian chemical industry. Romania is one of the largest  EU-member states,  the Romanian chemical industry having a tradition in the production of chemical products, especially oil and natural gas. Thus, this research attempts to fill this gap by using a primary data from a questionnaire that covers the entire transition period.

Basically, the study is constructed so that it will provide a list of the main strengths and weaknesses of Romania, which would influence a foreign investor to choose the proper location for a future investment when developing his strategy.

The data could be used by the authorities in order to develop strategies for atractting foreign investors in the Romanian chemical industry.

In our next papers we would take in consideration a comparative analysis among countries at European level or East-European level.